# Sustainability of Vertical Farming in Comparison with Conventional Farming: A Case Study in Miyagi Prefecture, Japan, on Nitrogen and Phosphorus Footprint

Jiarui Liu [1,*], Azusa Oita [2], Kentaro Hayashi [2,3] and Kazuyo Matsubae [1]

1   Graduate School of Environmental Studies, Tohoku University, 468-1 Aoba, Aramaki, Aoba-ku, Sendai 980-8572, Japan; kazuyo.matsubae.a2@tohoku.ac.jp
2   Institute for Agro-Environmental Sciences, National Agriculture and Food Research Organization (NARO), 3-1-3 Kannondai, Tsukuba 305-8604, Japan; a.oita@affrc.go.jp (A.O.); kentaroh@affrc.go.jp (K.H.)
3   Research Institute for Humanity and Nature, 457-4, Motoyama, Kamigamo, Kita-ku, Kyoto 603-8047, Japan
*   Correspondence: liu.jiarui.r7@dc.tohoku.ac.jp; Tel.: +81-22-752-2265

**Abstract:** The reduced requirement for nutrients in vertical farming (VF) implies that the potential for lower environmental impact is greater in VF than in conventional farming. In this study, the environmental impacts of VF were evaluated based on a case study of VF for vegetables in Miyagi Prefecture in Japan, where VF has been utilized in post-disaster relief operations in the wake of the 2011 Great East Japan Earthquake. The nitrogen (N) and phosphorus (P) footprints of these VFs were determined and analyzed to quantify the potential reduction in N and P emissions. First, the N and P footprints in conventional farming were calculated. Then, those footprints were compared with three different scenarios with different ratios for food imports, which equate to different levels of food self-sufficiency. The results show a decrease in the N and P footprints with increased prefectural self-sufficiency due to the introduction of VF. In addition to reducing the risks to food supply by reducing the dependence on imports and the environmental impacts of agriculture, further analysis reveals that VF is suitable for use in many scenarios around the world to reliably provide food to local communities. Its low vulnerability to natural disasters makes VF well suited to places most at risk from climate change anomalies.

**Keywords:** vertical farming; nitrogen footprint; phosphorus footprint; regional development; climate change adaptation policy; food self-sufficiency

## 1. Introduction

### 1.1. Importance of Nutrient Management

Nutrient input and water use for crop production result in the environmental pollution of aquatic ecosystems. One of the most studied environmental pollution problems is eutrophication, which occurs in water bodies due to excess nitrogen (N) and phosphorus (P) [1,2]. In the 21st century, one of the largest global challenges is to continuously increase crop production to ensure adequate food supply for the growing population while protecting the environment. In order to achieve this goal, it is essential to improve the nutrient use practices in agriculture, with particular emphasis on N and P [3–5]. For the conservation of aquatic ecosystems and food systems, it is important to manage nutrient inputs and outputs and reduce nutrient loss in production from agricultural systems by an integrated assessment based on life cycle processes [6,7].

### 1.2. Vertical Farming as an Emerging Technology in Agriculture

Vertical farming (VF) is an indoor method of growing crops with a controlled nutrient solution and recycled water in several layers with stable productivity (e.g., plant factories) [8–12]. The crops productivity of VF is higher than in conventional farming, and

the growth cycle is also faster [13–15]. As indoor farms, the benefits of VF are a lower requirement for water and pesticides and also the absence of fertilizer runoff in hydroponic systems [12]. However, limited crop species are suitable for commercial production using VF: the vegetables, fruits, herbs, and horticultural plants suited for VF have been discussed in earlier studies [14,15]. In addition to allowing year-round and stable crop production, VF also optimizes plant growth with lower environmental impacts. In VF, the water used is cycled back for reuse by returning it to the water tank [12,16], and the N and P contents in the nutrient solution are monitored by electrical conductivity [17]. The nutrient management system adopted in VF features an automated fertilization process, and only the necessary nutrients are provided for optimal growth: that is, there are no N and P runoff emissions with this managed water cycling [8,12,18,19]. VF has emerged as a potential alternative to conventional farming methods for the suitable crop species [20] and also has a potential to mitigate GHG emissions due to the shorter transport distances involved in the distribution of the produce, which makes it a more sustainable approach to agriculture [21–23].

It is increasingly common to find VF in various countries around the world, especially in the United States, Western Europe, and Asia [22,24–26]. In Indonesia, VF has been adopted to address the shortage of farmland in urban areas [27] and was introduced as a post-tsunami measure in Aceh due to the earthquake in 2004 [28]. A case study in Sweden suggests that the large environmental impact of conventional farming can be reduced significantly by substituting conventional farming methods with VF techniques [29]. In the Philippines, VF is an acceptable farming option for onion cropping [10]. Another study showed that VF can mitigate the climate change impacts associated with conventional farming methods [30]. In arid regions, such as the countries of the Gulf Cooperation Council, including Kuwait, Bahrain, Qatar, the United Arab Emirates, Oman, and Saudi Arabia, or many parts of Africa, rapid population growth, tough climate conditions, and the lack of water resources present serious food supply risks: VF has become a key factor in adapting to climate change and reducing the food supply risks [31,32]. An example is Kenya, where groundwater has been exhausted due to the impacts of desertification. The potential of VF in Kenya is being explored through a partnership between the Kenyan government and the Association for Vertical Farming.

Japan is subject to natural disasters such as typhoons, floods, and tsunamis [33]. Among the 47 prefectures in total in Japan, Miyagi Prefecture was severely affected by the tsunami and nuclear accident that followed the Great East Japan Earthquake in 2011. Over 15,000 ha of farmland was significantly damaged [34] (about 12% of the total farmland in Miyagi Prefecture [35]), and the reuse of the disaster area as farmland is difficult due to contamination by sea salt or radioisotopes. Since its beginnings with strawberries, VF was adopted as a regional rehabilitation project to rebuild agriculture in the area [36]. In other words, VF was promoted as a way to compensate the local farmers who had lost their land, rejuvenate the economy, and also to provide locally sourced produce to the residents of that area. As a part of the "Tohoku reconstruction" strategies, a VF project was begun in 2014 in Ishinomaki in Miyagi Prefecture, with paddy and chrysanthemum production, while another VF project was focused on cultivating wheat and soybeans from 2014 in Natori in Miyagi Prefecture [37]. Furthermore, the world's largest artificial light VF with LEDs was established in Tagajo, in Miyagi Prefecture, where 10,000 lettuce plants can be harvested per day [38]. The importance of locally produced food was highlighted in food supply in the period after the Great East Japan Earthquake in 2011 when supply was stopped due to damage to the roads. From this perspective, VF was promoted as a post-earthquake recovery measure and also as a way to mitigate the threat to food supply from future earthquakes or other disasters [39].

In conventional farming, the dependency on imports puts more pressure on global agricultural production and the environment [40–42]. Japan is heavily reliant on imports to meet food demand. The food self-sufficiency rate determined on a calorie basis in Japan is low, having decreased from 73% in 1965 to 38% in 2017, and that determined on a domestic production capacity decreased from 86% in 1965 to 66% in 2017 [43,44]. This is far below

that of most other developed countries: in Canada, self-sufficiency based on calories is 255%, and based on production capacity, it is 120%, while the numbers for Australia are 233% and 133% [45]. Countries, such as Japan, that rely on other countries to meet their food demand are effectively outsourcing great environmental impacts to those countries [46]. However, supply from abroad is not guaranteed: to ensure domestic food supply, some countries have reduced their food exports, which has resulted in increasing food prices and has affected the global food supply [47].

To address this situation, a plan, known as "The food, agriculture, and village basic plan in Japan" was proposed in 2018. This plan is based on the need to increase self-sufficiency based on caloric needs to 45%, and one of the goals is to increase the production capacity to 75% by 2030 [48]. Although the average import ratio of all kinds of vegetables in Japan was only 22% in 2018 [49–51], a survey of the local production and consumption of vegetables in all 47 prefectures of Japan reveals the distribution of vegetables was unbalanced [49,50]. For example, in 2018, Miyagi Prefecture was not able to meet the demand for certain vegetables without relying on supplies from outside the prefecture and imports from other countries [52,53]. To increase food self-sufficiency in Japan, it was proposed that domestic vegetable production should be expanded with the introduction of sustainable agriculture [54,55]. While domestic crop production increases by the introduction of VF, a higher level of food self-sufficiency relieves the dependency on imports. It is important to improve food self-sufficiency using sustainable agriculture. In order to ascertain the sustainability of VF on a long-term basis in terms of N and P, the indispensable nutrients for crop production [56,57], the N and P environmental emissions associated with VF need to be monitored.

The N and P footprints are defined as quantitative indicators of the total environmental emissions of N and P at a prefectural level or in certain areas based on consumption in a one-year period [58,59]. While the footprint concept considers the whole supply chain, it has been reported that crop cultivation is the largest contributor to N and P footprints [60]. The importance of confirming N and P emissions in agriculture using a footprint analysis to evaluate the sustainability of VF and provide data on how the agricultural environment is affected has been highlighted in earlier studies [58,59]. To date, little research of this nature has been conducted on VF in Japan. In addition, it was highlighted in a recent review paper on sustainable agricultural practices that N and P use efficiencies (NUE, PUE) should be determined in efforts to optimize nutrient use: these indicate the proportions of N and P that are absorbed and used by the plants from the total N and P inputs [61]. In other words, the challenge is to increase crop production while reducing environmental impacts and minimizing resource depletion due to agricultural demand by utilizing N and P more effectively and sustainably [62]. The NUE and PUE have been increasingly used as indicators to assess the nutrient balances of N and P in nutrient use practices [61–64].

### 1.3. Objective

Within the context of considering the environmental impacts of replacing imported vegetables with production by VF in Japan, the objective of this study was to quantify the extent of the reduction in the N and P footprints with VF as a replacement of conventional farming from the footprint perspective. The feasibility and effectiveness of VF is assessed for its ability to increase NUE, PUE, and food self-sufficiency; prevent water degradation; and stabilize crop production. The role of VF in disaster-resilient and post-disaster reconstruction is also discussed in areas not only damaged by the triple disaster of March 2011 in Miyagi Prefecture (earthquake, tsunami, and nuclear accident), but the results are also expected to be applicable to other areas of the world affected by natural disasters.

To achieve this objective, the trends in VF in Miyagi Prefecture were assessed. The first step was to conduct a survey to determine how widespread VF has become in post-disaster Miyagi Prefecture and to create a distribution map. Then by considering 36 different vegetables consumed in Miyagi Prefecture (strawberries, melon, and watermelon are classified as vegetables in Japan) [50], the extent of the reduction in the N and P footprints

for increasing food self-sufficiency by introduction of VF was quantified. In all, nine vegetables were chosen as target vegetables: these nine vegetables represented 22% of vegetable imports in Japan in 2018 (including frozen and processed products) [52]. The N and P footprints in conventional farming and VF were calculated based on consumption within the prefecture with a focus on crop cultivation both within and outside the prefecture, including abroad.

Here, "the prefectural self-sufficiency" of food in Miyagi Prefecture is defined as the proportion of food produced locally (that is, the crops produced within the prefecture) of the total prefectural consumption, whereas "the self-sufficiency" in the context of international trade is defined as the proportion of the domestic production (that is, the crops produced within Japan) of the total consumption. To evaluate the extent to which the N and P footprints were reduced in VF, a scenario analysis was conducted with changes in the dependencies of conventional farming and VF based on food self-sufficiency focusing on the nine target vegetables with relatively lower self-sufficiency at the national level.

## 2. Data and Methods

### 2.1. Data

2.1.1. Management of Vertical Farming in Japan

In 2020, over 85% of the tomato and strawberry market was represented by VF crops, while cucumbers, bell peppers, and asparagus grown in VF facilities represented between 60% and 70% of the market [50,65]. VF is becoming more widespread around Japan in recent years, but it is not possible for VF to replace conventional farming. At this point in time, the crop variety suitable for cultivation in VFs is severely limited, and VF techniques for crop production require further development.

There are roughly three types of VF in Japan: VF using natural light, VF using artificial lighting, and a combination of both. According to an annual survey in 2020 based on 2019 VF practices in Japan [66], a total of 164 factories used natural light, 187 used artificial lighting, and 35 used a combination of both. The number of factories using artificial lighting increased until 2015 and remained stable from 2015 to 2020, while those using natural light gradually increased [66].

2.1.2. Distribution of Vertical Farming in Miyagi Prefecture

Miyagi Prefecture, with an area of 7282 km$^2$ and population of 2,303,100 in 2018, is located in the Tohoku region in Japan (Figure 1). The six prefectures of the Tohoku region have a population of 8,842,610, and Sendai City, the capital of Miyagi Prefecture, is the largest city in this region with a population of approximately one million people (1,062,585 in 2018) [67].

There were 21 VF operators in Miyagi Prefecture in 2019 [66]. A total of 15 operators utilized natural light, 5 used artificial lighting, and 1 used both. The VF operators were concentrated in coastal areas, with two major areas, the surrounding area of Sendai City and the Yamamoto-cho area in the south of Miyagi Prefecture (Figure 1). Fourteen operators were established after the Great East Japan earthquake in 2011 (Table 1). The cultivated area per operator was more than 8000 m$^2$ for those using natural light, while those using artificial light used less than 5000 m$^2$. Considering the damage done to the soils of Miyagi Prefecture by the tsunami [68,69], the soilless nature of VF makes it highly suitable for agriculture in damaged areas. The uptake of VF has been supported by government subsidies, creating employment opportunities and helping with regional development [70,71].

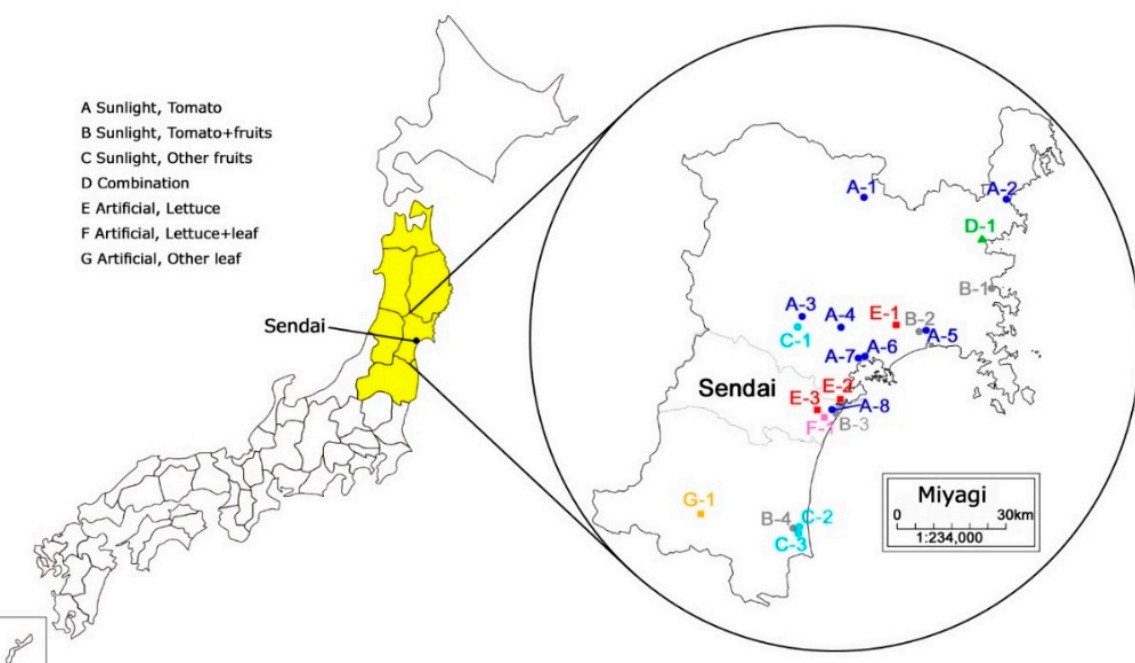

**Figure 1.** The distribution map of vertical farming in Miyagi Prefecture, Japan in 2019 (in the large circle). The yellow area is the Tohoku region.

**Table 1.** The different vertical farming scenarios in Miyagi Prefecture, Japan, in 2020.

| Type | No. | Start Year | Location | Cultivated Area (m²) | Production (t(10⁶ g) year⁻¹) |
|---|---|---|---|---|---|
| | A-1 | 2001 | Kurihara | 20,000 | 350 |
| | A-2 | 2016 | Kesennuma | 20,000 | 440 |
| | A-3 | 2005 | Kurokawa | 10,400 | 283 |
| | A-4 | 2017 | Kurokawa | 10,000 | 200 |
| | A-5 | 2012 | Ishinomaki | 15,700 | 200 |
| | A-6 | 2014 | Matsushima | 10,000 | 300 |
| | A-7 | 1996 | Matsushima | 10,000 | 300 |
| Natural light | A-8 | 2012 | Miyagino | 12,000 | * |
| | B-1 | 2016 | Ishinomaki | 24,000 | 630 |
| | B-2 | 2014 | Ishinomaki | 13,000 | 28.5 |
| | B-3 | 2012 | Miyagino | 28,000 | * |
| | B-4 | 2012 | Yamamoto | 14,030 | 178 |
| | C-1 | 2013 | Ohira | 18,000 | 315 |
| | C-2 | 2011 | Yamamoto | 21,600 | 10 |
| | C-3 | 2012 | Yamamoto | 8600 | 47.3 |
| Combination | D-1 | 2012 | Minamisanriku | 14,700 | * |
| | E-1 | 2010 | Ishinomaki | 330 | 27 |
| | E-2 | 2015 | Tagajou | 2300 | 292 |
| Artificial lighting | E-3 | 1988 | Wakabayashi | 600 | * |
| | F-1 | 2010 | Wakabayashi | * | * |
| | G-1 | 1989 | Shiroishi | 4100 | * |

* Data unknown.

## 2.2. Footprint Calculation

In this study, the conventional farming method was set as the current condition of agricultural production in 2018.

First, the loss of N and P in the production of the 36 vegetable crops mainly consumed in Japan was calculated (Table S1). Due to data limitations, we assumed that all the vegetables were grown by conventional farming in 2018. The prefectural-level N and P footprints were then estimated based on the amount of prefectural consumption of target vegetables,

including those grown and consumed in Miyagi Prefecture, grown in 46 other Japanese prefectures and consumed in Miyagi Prefecture, and grown overseas and consumed in Miyagi Prefecture. The loss of N and P in the production of imported vegetables, in Mg (i.e., $10^6$ g) N or Mg P loss per annual Mg production, were assumed to be the weighted average of the other Japanese prefectures. Three scenarios were developed with a focus on the 9 vegetable crops with imported ratios higher than the average imported ratios of the 36 vegetable crops in Japan in 2018, and the 9 vegetables were compared considering changes in the N and P footprints in conventional farming for various scenarios. To estimate the N and P footprints of 36 vegetables, 2018 data were used as the baseline. Nine vegetables with high import ratios in Japan were identified, and the scenarios described in Section 2.3 were developed with the assumption that various percentages of those vegetables were grown using VF rather than conventional farming.

The vegetable N footprint of Miyagi Prefecture was calculated using Equations (1)–(5) and the vegetable P footprint of Miyagi Prefecture was calculated similarly with an adjustment for the differences in the chemical nature of N and P, explained after Equation (5).

$$F_{j\alpha} = F_{\text{PP}j\alpha} + F_{\text{DIM}j\alpha} \tag{1}$$

In Equation (1), $F_{\text{PP}}$ is a one-year N footprint of the crop $j$ produced within the prefecture $\alpha$ and $F_{\text{DIM}}$ is a one-year N footprint of the crop $j$ imported from outside the prefecture $\alpha$ including transported from other prefectures and international import from overseas defined as "domestic import". Here, $F_{\text{PP}j\alpha}$ is defined as in Equation (2),

$$F_{\text{PP}j\alpha} = \frac{L_{j\alpha} C_{\text{PP}j\alpha}}{Q_{j\alpha}} \tag{2}$$

where $L$ is the loss of N in production (Mg N year$^{-1}$), $Q$ is the prefectural production amount (Mg year$^{-1}$) taken from the Statistical Survey on Crops [50], and $C_{\text{PP}}$ is the local consumption of crop $j$ produced locally in prefecture $\alpha$ (Mg year$^{-1}$) taken from a wholesale market survey [53].

Assuming that N fertilizer input ratios were as recommended by prefectural governments, $L$ of crop $j$ in the target prefecture $\alpha$ is calculated by subtracting harvested N, N taken out of field, and N plowed into soils with residue from the total N input by fertilizer, as in Equation (3),

$$L_{j\alpha} = (f_{\text{Chem}j\alpha} + f_{\text{Org}j\alpha}) S_{j\alpha} - (c_{\text{H}j}Q_{j\alpha} + c_{\text{R}j}Q_{j\alpha}w_j(1 - b_j\mu_j)) \tag{3}$$

where $f_{\text{Chem}}$ and $f_{\text{Org}}$ are the chemical fertilizer and the organic fertilizer applied per unit area ($10^4$ g N ha$^{-1}$) [72], $S$ is the area cultivated (ha) taken from the Statistical Survey on Crops [50], and ($f_{\text{Chem}} + f_{\text{Org}}$) $\times$ $S$ is the fertilizing amount as input. $C_{\text{H}}$ and $c_{\text{R}}$ are the ratios of N content in the harvested product taken from government reports and other literature [73–75] and residue taken from the National Greenhouse Gas Inventory Report [76], respectively, and $b$ is the fraction of the area that is burnt on a field, $\mu$ is the combustion factor [76], $w$ is the rate of residue to production [75,77]. Here, $c_{\text{R}} \times Q \times w \times b \times \mu$ is the N amount in burned residue counted as the loss of N in production [78], and $c_{\text{R}} \times Q \times w \times (1 - b\mu)$ is N taken out of field or plowed into soils as non-burned residue. The N amount plowed into soils with residue was also calculated as utilization. The N input by fertilizers was assumed to either go to harvested crops or residues or be directly lost to the environment.

Supposing that the consumption ratio of imported commodities in target prefecture $\alpha$ is the same as it is at the national level, $C_{\text{IMPORT}j\alpha}$, the consumption of imports in target prefecture $\alpha$, can be defined as follows,

$$C_{\text{IMPORT}j\alpha} = \frac{C_{\text{DOMESTIC}j\alpha}}{C_{\text{DOMESTIC}j}} \times C_{\text{IMPORT}j} \tag{4}$$

where $C_{\text{DOMESTIC}j\alpha}$ is the consumption of "domestically-produced" crop $j$ in the target prefecture $\alpha$, which is produced domestically. $C_{\text{DOMESTIC}j}$ is the national consumption of crop $j$, while $C_{\text{IMPORT}j}$ is the national import of crop $j$.

Supposing the N loss per production is the same as the national average for the transported from outside Miyagi Prefecture and imported commodities, $F_{\text{DIM}}$ can be expressed by Equation (5),

$$F_{\text{DIM}j\alpha} = \sum_{k=1}^{46} \frac{L_{jk} C_{\text{DIM}j\alpha}}{Q_{jk}} = \sum_{k=1}^{46} \frac{L_{jk}\left(C_{\text{TRANS}j\alpha} + C_{\text{IMPORT}j\alpha}\right)}{Q_{jk}} \tag{5}$$

where $L$ and $Q$ are based on the data of prefecture $k$, which are 46 prefectures other than Miyagi Prefecture in Japan. $C_{\text{TRANS}}$ and $C_{\text{IMPORT}}$ are the consumption of crop $j$ in Miyagi Prefecture, which is transported from production in other prefectures and overseas, respectively. Due to the limitation of data from each import country, the N footprint from import was calculated from the average of that from the other prefectures. On this basis, the vegetable N footprint of Miyagi Prefecture was calculated based on consumption data [53] and population statistics in 2018 [67]. The footprints for Miyagi Prefecture were estimated by multiplying the population of the Miyagi Prefecture by the averages of the per capita footprints of Tohoku region and Sendai city due to the limitation of the consumption data. Note that the footprints for strawberries, watermelons, and melons were calculated based on Sendai city only in this study.

The above method was used for the calculation of the N footprint and the P footprint of Miyagi Prefecture was also calculated according to Equations (1)–(5), but the value of $\mu$ was set at $\mu = 0$ for P in Equation (3) because there is no volatilized P in burned residue [79]. In the calculation of the N and P footprints, none of the crops planted that would fix N, such as legumes or alfalfa, were considered. These were the limitations of the methodology in this study.

### 2.3. Comparison Analysis

In order to verify the extent to which the N and P footprints are reduced due to the wider introduction of VF in Miyagi Prefecture, three scenarios at different import ratios were established. These scenarios assumed the international imported vegetables were substituted with the vegetables produced locally by VF. Scenario 1 is to halve the import ratios of the target vegetables in 2018. Scenario 2 is to halve the import quantities of the target vegetables in 2018. Scenario 3 is to have all target vegetables produced domestically.

As the import quantities and ratios of each vegetable show in Table 2, the targeted nine types of vegetables with an import ratio of over 22% were divided into three groups: currently VF grown (grown extensively in VF in 2018), possibly VF grown (planted in conventional farms mainly but possibly grown in VF, e.g., lettuce in Japan), and potentially VF grown (potentially grown in VF with a high risk of failure due to insufficient social and economic acceptance [80–84]). Due to the limitation of the consumption data, the import ratios of vegetables in Miyagi Prefecture were assumed to be the same as the ratios for the entire Japan, and the scenarios and vegetables groups were established based on the assumed import ratios. The N and P footprints on scenarios were calculated through Equations (1)–(5). The nutrient solution is recycled with controlled water cycling and does not run off [18]. Therefore, it is reasonable to assume that the N and P losses in VF are negligible. Then, the differences in the N and P footprints for conventional farming and VF were determined for Miyagi Prefecture.

**Table 2.** Quantities and ratios of import in Japan for each target vegetable crop in 2018 and for the three scenarios *.

| Vegetable Groups and Crops | Current in 2018 | | Scenario 1 | | Scenario 2 | | Scenario 3 |
|---|---|---|---|---|---|---|---|
| | Quantity $10^9$ g | Ratio % | Quantity $10^9$ g | Ratio % | Quantity $10^9$ g | Ratio % | Ratio % |
| Currently VF grown | | | | | | | |
| Tomatoes | 260 | 43 | 94 | 22 | 130 | 28 | 0 |
| Bell peppers | 41 | 28 | 19 | 14 | 20 | 16 | 0 |
| Possibly VF grown | | | | | | | |
| Spinach | 52 | 40 | 19 | 20 | 26 | 25 | 0 |
| Celery | 8.0 | 24 | 3.5 | 12 | 4.0 | 14 | 0 |
| Asparagus | 12 | 42 | 4.2 | 21 | 5.8 | 28 | 0 |
| Broccoli | 75 | 42 | 28 | 21 | 38 | 26 | 0 |
| Welsh onion | 67 | 27 | 28 | 14 | 34 | 16 | 0 |
| Potentially VF grown | | | | | | | |
| Pumpkin | 103 | 58 | 31 | 29 | 52 | 41 | 0 |
| Melon | 27 | 35 | 11 | 18 | 14 | 22 | 0 |

* The scenarios analyzed were as follows: (1) import ratios of the target vegetables become half of the ratio in 2018; (2) import quantities of vegetables become half of the quantity in 2018; (3) all target vegetables are domestically produced.

## 3. Results

### 3.1. Prefectural-Level N and P Footprints

The N and P footprints of all 36 investigated vegetables for conventional farming in Miyagi Prefecture were calculated in this study. The total N footprint of the vegetables was 3119 Mg N year$^{-1}$, while the total P footprint was 626 Mg P year$^{-1}$. The proportional footprint of the nine vegetables we propose could be primarily grown in VF was 32% each for both N and P. The results of the nine target vegetables in the scenario analysis is shown in Table 3. In the conditions of 2018, the total N and P footprints were 992 Mg N year$^{-1}$ and 198 Mg P year$^{-1}$, respectively, while the proportion of the possibly VF grown group (such as Welsh onions) was over 60% of the total N and P footprints. The trends of the N footprints for each vegetable were similar to those of the P footprints. Among the target vegetables, Welsh onions accounted for the highest N and P footprints, at 238 Mg N year$^{-1}$ and 58 Mg P year$^{-1}$, whereas celery accounted for the lowest, at 9.4 Mg N year$^{-1}$ and 2.4 Mg P year$^{-1}$, respectively. These results reveal great differences in the N and P footprints of each vegetable.

**Table 3.** The nitrogen and phosphorus footprints of target vegetables for the different production scenarios * in Miyagi Prefecture, Japan, compared with the production scenario in 2018.

| Vegetables | N Footprint (Mg N year$^{-1}$) | | | | P Footprint (Mg P year$^{-1}$) | | | |
|---|---|---|---|---|---|---|---|---|
| | Current (in 2018) | Scenario 1 | Scenario 2 | Scenario 3 | Current (in 2018) | Scenario 1 | Scenario 2 | Scenario 3 |
| **Total** | **992** | **758** | **808** | **624** | **198** | **153** | **163** | **127** |
| **Currently VF grown** | **163** | **126** | **133** | **103** | **34** | **26** | **28** | **21** |
| Tomatoes | 113 | 84 | 90 | 67 | 25 | 18 | 20 | 15 |
| Bell peppers | 50 | 42 | 43 | 36 | 9.0 | 7.5 | 7.7 | 6.5 |
| **Possibly VF grown** | **607** | **483** | **507** | **407** | **128** | **103** | **107** | **87** |
| Spinach | 157 | 119 | 126 | 96 | 28 | 21 | 22 | 16 |
| Celery | 9.4 | 8.2 | 8.3 | 7.2 | 2.41 | 2.09 | 2.13 | 1.84 |
| Asparagus | 54 | 39 | 42 | 30 | 9.2 | 6.6 | 7.2 | 5.2 |
| Broccoli | 148 | 110 | 118 | 88 | 30 | 22 | 24 | 18 |
| Welsh onions | 238 | 207 | 211 | 184 | 58 | 51 | 52 | 46 |
| **Potentially VF grown** | **222** | **149** | **168** | **114** | **37** | **25** | **28** | **19** |
| Pumpkin | 145 | 88 | 104 | 64 | 22 | 13 | 16 | 10 |
| Melons | 78 | 61 | 64 | 50 | 14.7 | 11.6 | 12.1 | 9.5 |

* The scenarios analyzed were as follows: (1) import ratios of the target vegetables become half of the ratio in 2018; (2) import quantities of vegetables become half of the quantity in 2018; (3) all target vegetables are domestically produced.

The total of the per capita N and P footprints of the nine target vegetables were 431 g N capita$^{-1}$ year$^{-1}$ and 86 g P capita$^{-1}$ year$^{-1}$, respectively, in 2018 (Figure 2). The per capita N footprints were 71 g N capita$^{-1}$ year$^{-1}$, 264 g N capita$^{-1}$ year$^{-1}$, and 97 g N capita$^{-1}$ year$^{-1}$ in the currently VF grown (such as tomatoes), possibly VF grown, and potentially VF grown (such as pumpkins) groups, respectively, while the per capita P footprints were 15 g P capita$^{-1}$ year$^{-1}$, 55 g P capita$^{-1}$ year$^{-1}$, and 16 g P capita$^{-1}$ year$^{-1}$. While the per capita N and P footprints of each vegetable exhibited similar trends between their total N and P footprints, these differed greatly between different kinds of vegetables.

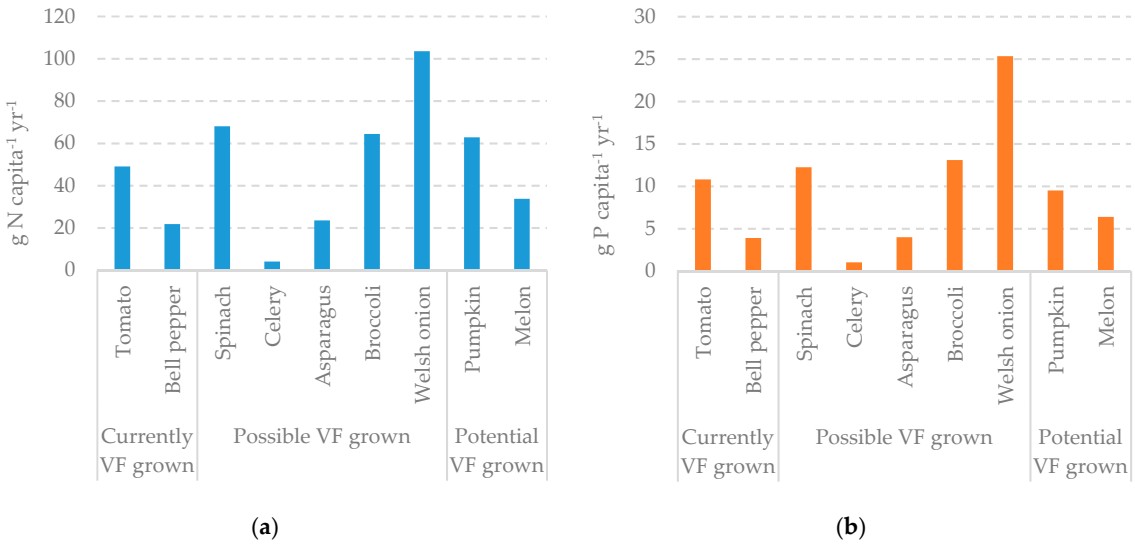

(**a**)                                              (**b**)

**Figure 2.** The current per capita vegetable footprints in Miyagi Prefecture, Japan, in 2018: (**a**) per capita nitrogen footprints; (**b**) per capita phosphorus footprints.

### 3.2. Results of the Scenario Analysis

The total N and P footprints of nine target vegetables in Miyagi Prefecture reduced by 234 Mg N year$^{-1}$ (24%) and 45 Mg P year$^{-1}$ (22%) in scenario 1, with import ratios half of the ratio in 2018; by 184 Mg N year$^{-1}$ (19%) and 35 Mg P year$^{-1}$ (18%) in scenario 2, with

import quantities half of the quantity in 2018; and by 368 Mg N year$^{-1}$ (37%) and 71 Mg P year$^{-1}$ (36%) in scenario 3, with all target vegetables domestically produced (Table 3 and Figure 3a).

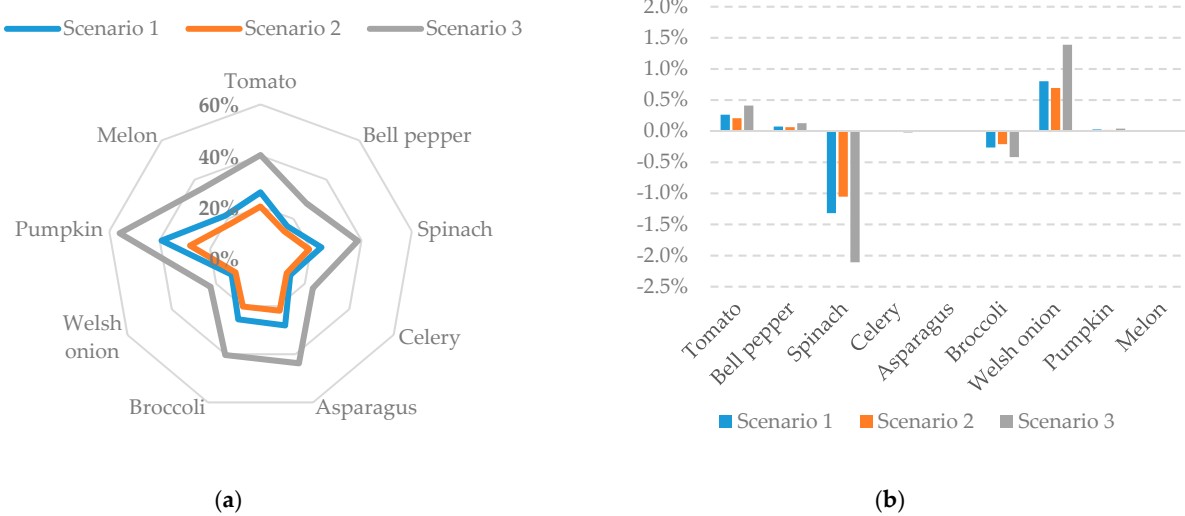

(**a**)  (**b**)

**Figure 3.** The reduction ratio in the nitrogen and phosphorus footprints of each vegetable for different production scenarios in Miyagi Prefecture: (**a**) the reduction ratio of nitrogen footprints; (**b**) a comparison of the reduction ratios of the phosphorus and nitrogen footprints. The scenarios analyzed were as follows: (1) import ratios of the target vegetables become half of the ratio in 2018; (2) import quantities of vegetables become half of the quantity in 2018; (3) all target vegetables are domestically produced.

By introducing VF to substitute the imported vegetables, it is possible to reduce the total N and P footprints by more than 35%. The reduction ratios of the N and P footprints of each vegetable were similar in the same scenario. The reduction ratio of the footprints for pumpkins was the highest, whereas the lowest was for Welsh onions in each scenario. Compared to the N footprint reduction, the reduction ratios for P footprint were over 1% higher for spinach and over 0.7% lower for Welsh onions (Figure 3b).

Among three vegetable groups, the reduction in the N and P footprints in the possibly VF grown group, such as Welsh onions, was the largest, whereas the reduction ratio in this group was the lowest in each scenario. The reduction ratio in the footprint of the potentially VF grown group was the highest in each scenario. This reveals that the potentially VF grown group has a better reduction effect due to large reduction of the N and P footprints for pumpkins and melons. The reduction effect of N and P footprints in the possibly VF grown group was the lowest.

### 3.3. Results for N and P Use Efficiency

As shown in Figure 4, in conventional farming, the NUE was the highest for broccoli (30%) and the lowest for pumpkin (5%), while the PUE was the highest for melons (23%) and the lowest for asparagus (4%). By introducing VF, NUEs for each vegetable increased to 30–60% in scenario 1, 28–57% in scenario 2, and 41–72% in scenario 3 (Figure 4a). While PUEs increased to 27–53% in scenario 1, 25–50% in scenario 2, and 39–68% in scenario 3 (Figure 4b). This reveals the NUE and PUE increased in each scenario on introducing VF instead of importing food, and there was significant difference compared with NUE and PUE for each vegetable. The NUE for bell peppers, celery, asparagus, broccoli, and Welsh onions was higher than the PUE for these vegetables, whereas that for pumpkin and melons was significantly lower than the PUE in each scenario.

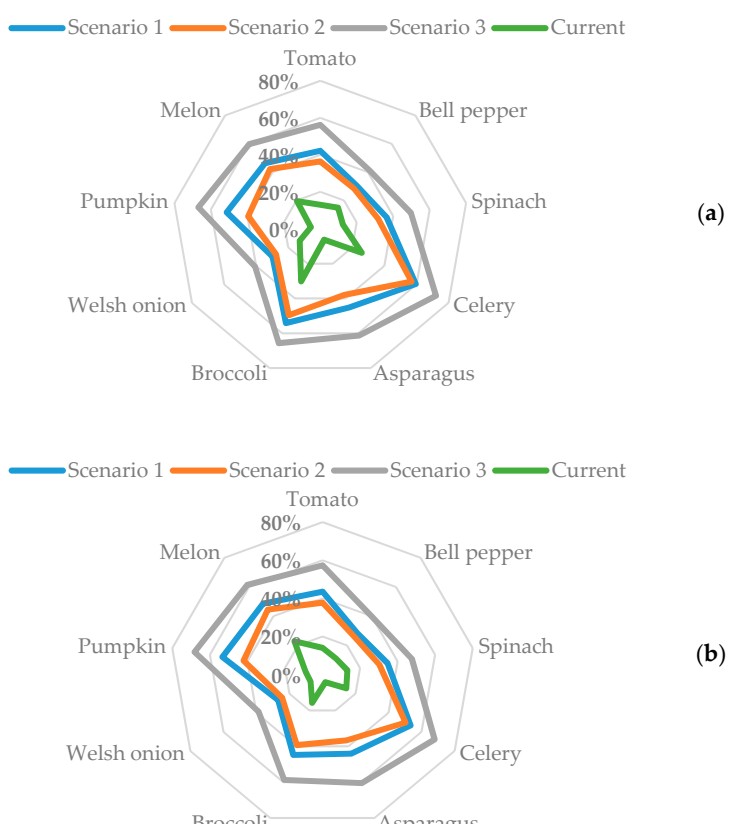

**Figure 4.** The nutrient use efficiencies for different scenarios in Miyagi Prefecture: (**a**) nitrogen use efficiency; (**b**) phosphorus use efficiency. The scenarios analyzed were as follows: (1) import ratios of the target vegetables become half of the ratio in 2018; (2) import quantities of vegetables become half of the quantity in 2018; (3) all target vegetables are domestically produced.

## 4. Discussion

The contributions of the reductions in the N and P footprints by introducing VF were considered in terms of the following: the water environment, food self-sufficiency, and disaster-resilient agriculture.

### 4.1. Impact of VF on Averting the Risk of Water Degradation

One of the environmental impacts of agriculture is the pollution caused by the N and P losses [85]. In conventional farming, N can be lost to the environment in forms of $N_2O$, $NO_3^-$, or $NH_3$ [86,87]. $NH_3$ is released to the atmosphere with volatilization [78], and $NO_3^-$ goes to waterways as leaching or runoff [86,88]. $N_2O$ is a significant GHG [3]. $N_2O$ emissions in agriculture are significant due to the utilization of fertilizers and manure [86]. Together with the P compounds discharged in the dissolved and particulate forms by runoff [78], N lost to the environment contributes to the eutrophication of aquatic ecosystems [3,89–92]. One important factor in reducing the N and P loss in production to the environment in VF is the reuse of the nutrient solution. A nutrient solution including N and P is recycled in VF by controlling the nutrient composition rather than discharging the nutrients into the environment. This study confirmed that the total N and P footprint of the target vegetables of conventional farming in Miyagi Prefecture were 992 Mg N year$^{-1}$ and 198 Mg P year$^{-1}$ in 2018, and that VF effectively reduced N emissions by 368 Mg N year$^{-1}$ (37%) and P emissions by 71 Mg P year$^{-1}$ (36%) (Table 3). This means that the N and P loss in the production of food to be consumed in Miyagi Prefecture in remote places can be reduced by introducing more VF as well as keeping the environment of the local production areas within Miyagi Prefecture, protecting both local and remote ecosystems [93]. The study of reducing N and P footprints by VF provides a reference for N and P emission

standards in agriculture. However, the actual application ratios of chemical and organic fertilizers in agricultural production may be higher than the recommended application ratios provided by prefectural governments used in this article.

According to the results of this study, the nutrient use efficiencies of conventional farming were low, from 5% to 30% for the NUEs and from 4% to 23% for the PUEs, whereas they can be increased by introducing VF to 41–72% for the NUEs and 39–68% for the PUEs. In earlier studies, the NUE for duckweed in the hydroponics system was increased to 67% from 25% in conventional farming, while the PUE in the hydroponics system was 33% [94,95]. Those for water hyacinth increased to 63% for total N and 79% for total P [94]. Clearly, VF contributes to improving the quality of water by removing N and P from runoff and because there is no leaching water, unlike in conventional agriculture. Therefore, VF is an effective approach to agriculture with a mitigated risk of water-quality degradation.

### 4.2. Impact of VF on Food Self-Sufficiency and Urban Agriculture

The estimated prefectural food self-sufficiencies of the nine target vegetables ranged from 1% (melons) to 83% (Welsh onions) in conventional farming and can be increased by 27–111% by introducing VF. The results show that VF is an option to reduce N and P footprints and promote self-sufficiency for major food importers such as Japan.

It is foreseeable that exclusive reliance on improving conventional farming to ensure food security will one day change due to resource shortage caused by rapid urban expansion and industrial development [96]. As shown in this study, VF, as a type of urban agriculture, allows for food cultivation in areas where farmland is scarce or damaged. VF is a sustainable urban agricultural technique with great potential to improve self-sufficiency in countries lacking in farmland or with barren land [97]. In Singapore, VF was promoted in a new policy in 2019 designed to promote an improvement in self-sufficiency from 10% to 30% by 2030 [98]. In another case study of Lyon, France, the positive environmental and social benefits of VF were highlighted by the increase in self-sufficiency and improved adaptability of the city [99]. In the United States, several VF facilities have been established in Chicago, and the world's largest VF facility is in New Jersey [100]. VF has also become more common in other countries such as Italy and Brazil. Because of the characteristics of soilless cultivation, VF is a viable option in countries with insufficient farmland and also in regions that cannot engage in conventional farming due to the limitations of topography and poor soil fertility. These studies point to a trend that VF will gradually replace conventional farming in urban areas in the future, in effect contributing to higher self-sufficiency and a reduction in land use for agricultural purposes [93].

### 4.3. Potential of VF as a Disaster-Resilient Agriculture

Natural disasters such as landslides, heavy rainfalls, and floods, particularly in the rainy season and the typhoon season, threaten the food security of Japan. In the past decade, there have been 10 earthquakes with a magnitude of over 6.0 in Japan, including the massive Great East Japan Earthquake in 2011 [101]. Furthermore, persistent rainstorms and super typhoons have become increasingly common in the years from 2012 to 2020 [101]. The damage to agriculture from massive rainfall events, typhoons, and violent earthquakes in 2018 was estimated at JPY 568 billion (USD 5 billion) including JPY 112 billion (USD 1 billion) in crop production, and it was the second worst year in the decade after 2011 [68]. In another survey, it was reported that the total damage to crop production in the 25 prefectures affected by unseasonably heavy rainstorms in July 2020 was JPY 1.4 billion (USD 12 million) [102]. As global warming accelerates, natural disasters are likely to become more frequent and intense globally. According to an international disaster database, the number of annual disasters in 2018 in developing Asia and the Japan region was the highest since records began in the 1970s [101].

The widespread application of VF is considered a way to offset the damage caused by disaster to the food supply. According to the survey results, which were discussed in Section 2.1.2, the post-disaster promotion of VF was revealed. The main advantage of VF is

that the crops are grown completely indoors and are, therefore, unaffected by rain, drought, and most other natural disasters, and the cultivation conditions are controlled [11]. From this perspective, VF can be considered disaster resilient [33]. However, it is not reasonable to claim that VF is disaster-proof since disruption to the electricity and water supply due to a violent earthquake or a flood would pose an immediate risk to VF, with both time and cost required for recovery [33]. The risk posed by violent disasters to VF needs to be assessed and efforts to mitigate the potential damage need to be considered. To be less susceptible to disaster, a back-up generator system would enable VF production to continue in the event of electricity outages, for example.

After the Great East Japan Earthquake in 2011, VF was widely adopted in agricultural reconstruction efforts in the areas damaged by the earthquake and tsunami. As a response to the devastation of regional agriculture due to these multiple concurrent natural disasters, it was essential to restore agricultural capacity with no further environmental impacts. Based on the results in Section 2.1.2, 14 of the 21 VF operators in Miyagi Prefecture began their operations in the period from 2011 to 2017. This uptake in VF was at almost twice the pace outlined in the Tohoku reconstruction strategy, and the operators have been in business for at least 5 years. The extensive implementation of VF in Miyagi Prefecture provides the opportunity to investigate the specific impacts of VF as a form of agricultural reconstruction. The results of this study confirm the suitability and stability of VF for post-disaster agriculture from the perspective of reducing the N and P footprints and also for its potential to restore agricultural capacity.

The suitability of VF as a system to provide locally grown food with a reliable high production rate, with high efficiency, and without occupying farmland has been demonstrated in several studies [94,95,103–105]. It has the potential to be used anywhere, and planting can be done at any time regardless of the location of the VF facility or the season [8,106,107]. For example, in Bangladesh, where cyclones occur frequently, an adaptation will be implemented to both increase productivity and reduce the risks posed by natural disasters to conventional farming [108]. Similar to the case in Japan, in Aceh, Indonesia, which was damaged by the earthquake and tsunami in 2004, VF has been a central part of the post-disaster recovery program, since it is both disaster resilient and sensible for post-disaster scenarios [28]. This reveals that post-disaster agricultural development is a key factor in mitigating the impacts of natural disasters in the future. That is, VF is suitable not only for Miyagi Prefecture in Japan but should be considered a necessary new farm technology for use in scattered islands and other countries with frequent natural disasters, such as Indonesia and Bangladesh, or where food security needs to be improved. Considering the extreme risks to the food supply posed by natural disasters, VF has the potential to decrease the dependence on conventional farming and to accelerate the move toward more sustainable agriculture.

## 5. Conclusions

The N and P footprints of vegetables in VF and conventional agriculture for Miyagi Prefecture were compared based on the change in replacing imported vegetables with production from VF in Japan. In the case of VF, the footprints of the target vegetables were reduced. The N footprint was reduced by 37%, at 363 M g N year$^{-1}$, and the P footprint was reduced by 36%, at 71 Mg P year$^{-1}$. The results indicate that expanding the scale of production in VF has the potential to reduce pollution due to excessive N and P in the aquatic environment, to improve prefectural and even national self-sufficiency, and to prevent water quality decline while saving water resources. The vital role played by VF in the regional agricultural reconstruction of Miyagi Prefecture after the Great East Japan Earthquake in 2011 was also shown. Further analysis revealed that VF is well suited for use in disaster-prone regions in Japan and in other parts of the world. The data provided by this study have potential for use in the formulation of policies designed to reduce N and P emissions by the introduction of VF. In the future, this research can be expanded by conducting a life-cycle analysis of the environmental footprint and carbon emissions of VF

and comparing the results with those of conventional agriculture with agricultural imports taken into consideration.

**Supplementary Materials:** The following are available online at https://www.mdpi.com/article/10.3390/su14021042/s1, Table S1: The categories of 36 vegetables consumed in Miyagi Prefecture, Japan, produced via conventional farming, including 9 target vegetables whose import ratios were above average in Japan in 2018.

**Author Contributions:** Conceptualization, J.L., K.M. and A.O.; methodology, J.L., A.O. and K.H.; investigation, J.L. and A.O.; resources, J.L., A.O. and K.M.; writing—original draft preparation, J.L.; writing—review and editing, A.O., K.M. and K.H.; visualization, J.L.; supervision, K.M.; project administration, J.L.; funding acquisition, K.M. and A.O. All authors have read and agreed to the published version of the manuscript.

**Funding:** This work was supported in part by JSPS KAKENHI grant numbers JP17H00794 and JP19K20496.

**Institutional Review Board Statement:** Not applicable.

**Informed Consent Statement:** Not applicable.

**Data Availability Statement:** The data presented in this study are available in the references and supplementary material.

**Acknowledgments:** We would like to thank Green River Holdings, Inc., and De Liefde KITAKAMI, Inc., for their helpful introduction and discussions on VF.

**Conflicts of Interest:** The authors declare no conflict of interest.

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
