# Peer review of "Sustainability of Vertical Farming in Comparison with Conventional Farming: A Case Study in Miyagi Prefecture, Japan, on Nitrogen and Phosphorus Footprint"

_sustainability, doi:10.3390/su14021042_

Round 1
Reviewer 1 Report
Sustainability review
Sustainability of Vertical Farming in comparison with 2 conventional farming: A case study in Miyagi Prefecture, 3 Japan on Nitrogen and Phosphorus Footprint
Overall: I found this paper interesting as it relates to whether or not VF emits fewer nutrients into the environment than conventional farming and whether it has a lower environmental impact as calculated on a life cycle basis. I cannot assess whether work of this nature has been published before because I have not read or searched for similar articles. Although I found this an interesting question, I do not think this paper meets the objectives the authors state that they meet. One of the main problems is that the authors state that VF emits no N or P into the environment without any citations or data to support this contention. If you start with an unsupported claim like that, then of course it will reduce N and P emissions. This is in need of data or at least citations to support this contention.
Another key issue is that it is unclear what they are comparing. It reads as if they are comparing 9 vegetables that could be grown using VF, or perhaps the 9 plants currently being grown using VF to 36 vegetables grown using conventional farming. If so, this is not a legitimate comparison because it does not appear to have been normalized. Data sources for some calculations are unclear, particularly how N and P from imports was calculated. There is no recognition of the very different ways in which N and P interact chemically after being applied to crops, particularly the volatilization of N, which is not present with P.
Introduction: Good information about VF as an emerging technology, but I did not find it to be comprehensive enough to give a good background to the stated objectives of the paper. A lot of the information in the Discussion should be moved into the Introduction so that the reader has a better context for the objectives of the paper.
Objectives:
lines 76-77" the objective of this study was to determine the extent of the reduction in the N and P footprints with VF for use in policymaking on the expansion of VF from a life cycle assessment perspective."
The authors make the statement later in the paper that VF water is recycled and therefore, no N or P is emitted into the environment. There is no data or study cited to support this assertion. Additionally, they do not address this from a life cycle assessment, which would include comparing the infrastructure needed for VF (pumps, buildings, electricity or fuel use to heat or cool the buildings, etc.) to that of conventional farming. If they are defining life cycle assessment differently, they have not provided a different definition in the paper. I also was unable to find any discussion of policy-making, so I am unclear why that was included as an objective.
Lines 79-80 " Suggestions are then provided based on the data about N and P footprints for strategies to reconstruct agriculture in the areas damaged by the triple disaster of March 2011 in Miyagi Prefecture."
Several scenarios are included in tables increasing use of VF and estimating the reduction in N and P emitted to the environment, but I did not find any strategies for reconstructing agriculture in areas damaged by the earthquake/tsunami/radiation leak.
Lines 81-82 " Finally, the feasibility and effectiveness of VF in post-disaster agriculture is assessed for its ability to adapt and stabilize crop production in areas affected by natural disasters."
This objective was more closely addressed in the paper as the authors discuss how more VF crop production can be incorporated to increase food self-sufficiency in the region.
Lines 85-86 - the authors state that there are "36 vegetables of the Miyagi Prefecture." These are not listed. It is not clear whether these are 36 that are grown using VF, or the main vegetables grown by VF and conventional farming, or the main vegetables eaten in this region. This is more clear on lines 145 and 146, but needs to be stated clearly earlier in the paper.
Throughout the paper there are different numbers of vegetables mentioned. Table 1 lists tomatoes (which I assume are being classified as a vegetable in the paper although they are technically a fruit), lettuce, herbs, paprika (also confusing because paprika in English is a spice made from peppers - are the peppers used to make paprika grown using VF?), leaf vegetables, bell peppers, chrysanthemum (not a vegetable and is this for consumption or for flowers not to be consumed?), and strawberries (also not a vegetable). Table 2 lists 9 edible plants that are or could be grown using VF and these 9 are consistently used in tables for VF throughout, but the authors still refer to the 36 vegetables that they investigated. These are never identified. From the paper, it appears that the N and P footprints of these in conventional farming were calculated, as well as the footprints of imports. If this is the case, then clearly VF will have a lower N and P footprint than conventional farming because of a comparison of 36 vegetables to 9 grown by VF. What is and is not being compared in this paper is not clear.
Lines 112 - 142 - there is a substantial discussion of the use of LED versus natural lighting in VF in this section of the paper, but it is not mentioned later. This would be relevant if there was truly a life-cycle analysis of environmental impact or contribution to climate change, both of which the authors mention. However, there is not a life-cycle analysis for either purpose, so it is unclear what relevance use of LED versus natural light has. If LED increases production, that might merit a sentence or two, not several paragraphs.
Lines 121 - 122 " It is evident that VF with artificial lighting required less cultivated area than those which utilize natural light, and that crop production is higher." Why is this evident? The reader is given no information that would support this, or a citation to a paper that found it to be a fact. Although at lines 137 - 138 the paper says: "Fourteen operators were established after the Great East Japan earthquake in 2011 (Table 1). The cultivated area per operator was more than 8000 m2 for those using natural light while those using artificial light used less than 5000 m2." Calories produced by the vegetables grown under natural and artificial light would be more relevant, this just indicates that operators using natural light had a larger area in which they were growing vegetables. Also, since one type of vegetable (lettuce primarily) is being grown in artificial lighting and tomatoes in natural lighting, how is the comparison being made? Are you using kg production/m2? Are you making the assumption that 1 kg of lettuce is the equivalent of 1 kg of lettuce? It seems like calories would be a better indicator of that. Also, tomatoes are larger plants and need more room with less of the plant available for consumption compared to lettuce - most of which can be consumed. These issues are not considered or addressed.
lines 145 - 147 " First, the loss of N and P in production of 36 vegetable crops in conventional farming were calculated, then the prefectural level N and P footprints were estimated based on the amount of consumption of vegetables grown within and outside the target prefecture of Miyagi."
Why are vegetables grown outside the target prefecture used? Are they imported into Miyagi prefecture? Unclear why.
line 148 " Three scenarios were created to compare the prefectural footprints of N and P with the state in 2018." I believe what you are saying is that you used 2018 as your baseline for comparison. Reword the sentence to say "Three scenarios were created to compare prefecture footprints of N and P to the baseline of N and P emissions in 2018."
Again, are you comparing N and P emissions of the 36 vegetables grown using conventional farming and to what are you comparing them? To the plants currently grown using VF, to the 9 plants that could be grown by VF? There are places in the paper where that is not clear - and if you are comparing nutrient emissions from 36 vegetables to those from 9 it is not a legitimate comparison.
A list of the 36 vegetables that were investigated would be helpful to the reader.
There are problems with equation 3 lines 158- 165 " where fChem and fOrg are the chemical fertilizer and the organic fertilizer applied per unit area [104 g N ha−1_ _or 104 g P ha−1_] [45], S is the area cultivated [ha] taken from the Statistical Survey on Crops [37], cH and cR are the ratios of N or P content in the harvested product and residue, respectively (the contents ratios were primarily taken from the National Greenhouse Gas Inventory Report of Japan [46], while auxiliary data were taken from a number of other government reports [47-49]), and b is the ratio of burned area in the field, μ is the combustion factor, t is the ratio of residue taken out of the field excluding burned biomass [47] and w is the rate of residue to production [49, 50]. The value of μ is set at μ=0 for P. All the contained P remains in ash."
Are these terms: ?? and ?? the same geographic area? One cannot tell from your description of the terms or from the paper. If they are not, then the equation is not balanced and does not provide an accurate representation of loss and gain of nutrients.
There are several instances where the term "or" is used where I don't believe that is what is meant. For example line 160 " cH and cR are the ratios of N or P content" but from looking at the equation, I assume that one is for N content and the other for P content, so use of the word "or" is confusing and does not represent what you are trying to do in the equation.
What are auxiliary data? I have no idea what you mean by this term. The citations [47-49] are not very helpful in trying to determine what this auxiliary data is either.
There is no clear explanation why burned residue is treated as a removal of nutrients in the equation, particularly when you state in the paper that P remains in the ash left on the field. So, if it is left in the field, that would mean that it can contaminate water supplies and should not be subtracted.
What is a combustion factor - and how does this relate to nutrient retention or loss? From the last sentence of the paragraph, I am guessing that it might be volatilization of N??? If it is, it should only be multiplied by whichever term represents N, and not by the term representing P.
Why is this part of the equation: (????+(1−????)??) multiplied by the term ???????? and not also by this term: ?H????? Furthermore, why is ? only multiplied by the ?????? term?
Finally, the source for chemical and organic fertilizer [45] are recommendations for application. Are these recommendations enforced by a governmental agency or any other entity? How accurate are these numbers likely to be? If they are likely not accurate, it should at least be acknowledged in the paper.
Equation 5 - you are summing from k=1 to 46. I assume that these are the 46 prefectures in Japan excluding Miyagi. This is stated nowhere in the text for this equation, so a reader will wonder why you are summing from 1 to 46. It is not stated until lines 185 - 186 of the paper. You need to explain more clearly why you are using data over all of Japan rather than just in Miyagi Prefecture since earlier in the paper you state that you are restricting your analysis to Miyagi Prefecture. Data limitations are not stated clearly throughout the paper.
Section 2.3 - Comparison analysis
Lines 184 - 189 "Japan is heavily reliant on imports to meet food demand. Therefore, its self-sufficiency is low [51, 52]. A survey of the local production and consumption of vegetables in 47 prefectures of Japan [37, 53] reveals an unbalanced distribution of vegetables. As the production of vegetables such as Welsh onions or tomatoes in Miyagi Prefecture is low to meet the high local, domestic requirement and international import is necessary to supply requirements. The average import ratio of all 36 kinds of vegetables was 22% in 2018 [37, 53, 54]."
It is unclear whether the geographic area analyzed in this section is Miyagi Prefecture or the entirety of Japan. The caption to Table 2 implies it is all of Japan. This needs to be clearly stated, as well as an explanation why the focus has shifted.
lines 207 - 211 "According to nutrient management in VF, a nutrient solution is used as a supplement when the N or P content is too low: that is, N and P are recycled with the nutrient solution and are not discharged into the environment. Therefore, N and P losses in VF are negligible. In addition, the N and P use efficiency (NUE, PUE) are introduced, which are the proportion of N and P which are absorbed and used by the plants from the total N and P inputs."
This needs at least one citation. It is the basis for the entire comparison of VF N and P losses to N and P losses from conventional farming. Without a creditable citation, or actual data, for this statement, there is no support for the claims of this paper.
Lines 220 - 223 "The N and P footprints of all 36 investigated vegetables at conventional farming in Miyagi Prefecture are calculated in this study. The total N footprint of the vegetables is 3,041 Mg N yr-1, while the total P footprint is 633 Mg p yr-1. Among these vegetables, the proportion of footprints on the nine target vegetables were 32% for N and 31% for P."
Last sentence wording should be changed to: "The proportional footprint of the nine vegetables we propose could be primarily grown in VF were 32% for N and 31% for P."
Section 3.3 - this is the first time in the paper that NUE and PUE are discussed. There is no background for why these measurements are important and why they improve with the use of VF. This seems like an afterthought in the study. Either there should be a better explanation earlier in the paper and a better explanation of its relevance to comparing conventional farming and VF, or this entire section should be taken out.
Discussion
Lines 275 - 277 "The contributions of the reductions in the N and P footprints by introducing VF were considered in terms of the following: the water environment, food self-sufficiency, post-disaster agriculture and climate change adaptation."
Was there ever a discussion or analysis of climate change adaptation? Although you say that it was considered, I think a statement that VF could help adapt to climate change without an analysis is very weak. I would say you made a statement, but not that you considered it.
lines 280 - 281 "The N and P losses in conventional farming seep into the soil mostly, and are discharged into the rivers and groundwater, where they can cause eutrophication [56]."
This statement is not true. Nitrogen and phosphorus react totally differently in soils. N is volatile and you have to apply a lot for it to seep into soils. N does NOT cause eutrophication. P does. This exhibits a shocking lack of understanding of the nutrients about which you are writing a paper. This lack of understanding is also reflected in equation 3.
lines 365-366 "In previous studies, it has been revealed that VF can be used anywhere and that planting can be done at any time regardless of the location of the VF facility [8]."
You are only citing one study, so the sentence should read "In a previous study . .."
lines 387-389 - "The potential role that VF can play in climate change mitigation by reducing greenhouse gas emissions due to the shorter transport distance [42], and in adaptation and enhancing climate resilience has been recognized [75]."
Without an analysis of whether the climate emissions from building infrastructure, pumping water, and power to heat and or cool VF is less than the carbon emissions of transporting vegetables (which would need to include how far they are being transported) this is a statement that is unsupported by facts.
Author Response
We would like to thank you for your suggestions and comments. These comments and suggestions were very useful in improving our manuscript.
Please see the attachment for the response to your comments.
Thank you very much again.

Reviewer 2 Report
General comments:
Overall, I had a great impression of the work, showing that VF can improve food production in urban areas, but also with positive feedback on the efficiency of using N and P. Currently, the paper seems enough in terms of innovation to the publication. In future work, I recommend including explicitly avoided C emissions over shorter transport distances. In addition, it would be worth discussing the effects of VF on local and remote (imported) ecosystem services, considering e.g. VF as Nature-Based Solutions (see, for example, https://doi.org/10.3390/resources10110109).
Specific comment: For figs. 2a,b and 3b present the grid-lines as dashed or dotted lines. For fig 3a remove the decimal in Y-axis numbers.
Author Response
We are happy that our paper met your approval. Thank you for taking the time to suggest to improve the manuscript.
Please see the attachment for the response to your comments.
Thank you very much again.

Reviewer 3 Report
- The title is somehow misleading suggesting straightforward sort of comparison with conventional farming. This applies also the first sentence of the Abstract: “Vertical Farming (VF) is a potential alternative to conventional farming methods”. Being more precise this story is about comparison of different structures of supply sources at the Miyagi Prefecture resulting with various N and P footprints, which differ between scenarios (table 3). In an indirect way the reduction of emissions is shown, but may be it would be worth to make the nature of the comparison clearer.2.
- Row 32. “Nutrient input and farmland use for crop production result in environmental pollution, like eutrophication, and land use competition”. Competition for land is another issue, I don’t see a reason to mix it with “pollutions”. If – it should be made clear what the Authors have in mind.
- Just for the clarity – does import means in this analysis goods imported to Japan from abroad or imported to the Prefecture from other parts of the country?
- I would suggest to re-phrase the formulation of the objective (row 76). Regarding “the context” – I would say it is not about “impacts of imported vegetables” but about impacts of replacing imports with the VF production. Secondly, there are two controversial elements:
- “for use in policy making on the expansion of VF” – in my opinion no need to indicate who and what use can make from the results of the research. I consider this is rather a statement for conclusions on policy implications.
- “from a life cycle assessment perspective” – in reality everything is part of the “life cycle”, however there is no formal LCA approach in the paper. This should be deleted.
- Row “…. the ratio of burned area in the field” – what is it about (burned?)
6. Row 203. Identical footnotes under tables 2 and 3 could be moved to the main text.
- Rows 214 and 255 – the same Table 3 - twice in these rows.
8. Row 334. Chapter 3 Impacts of VF on post-disaster agriculture
I am afraid the title of this chapter is not corresponding with the content. What’s more – it points to the very different problem. In my opinion this technology should be considered and approach reducing supply risk and a strategy to create a system resilient to potential disaster, not the “post-disaster” intervention. Much more appropriate is “VF has enormous potential as an adaptive policy to create sustainable agriculture” (row 374), although I would add word “more” before “sustainable agriculture” (fully, 100% sustainable cannot exist) and skip the word “enormous”.
Furthermore, bringing vegetables indoor leaves a certain amount of agricultural land potentially freed for other crops or set-aside. In both situations this land is still exposed to disasters.
- Row 376. Chapter 4 Impacts of VF on climate change adaptation
There are 2 aspects relevant – adaptation to climate change and/or reducing negative impacts on the climate, but not as it is in the title – “climate change adaptation”.
My main concern is, however, that this topic cannot be deeply discussed because it is hardly supported with the results of the analysis. The evidence is provided only for N and P efficiencies, while climate related impacts require much broader analyses, including energy use, inputs to construct VF facilities, transportation, as well as other uses of land. Possibly the LCA approach would be appropriate and may be such suggestion should be added to conclusions.
10. Row 414. “Further analysis revealed that VF is an adaptable type of architecture well-suited to post-disaster scenarios” – see comment 8. I would rather consider VF an “anti-disaster” approach.
Author Response
We thank you for your comments and suggestions. Thanks to your careful review of our manuscript, we feel the quality of the manuscript has improved.
Please see the attachment for the response to your comments.
Thank you very much again.

Round 2
Reviewer 1 Report
Line 36 should be "In order to achieve" not "On order to achieve"
line 54 - thank you for providing a citation for your assertion that there are no emissions of nutrients from VF, but Scientific American is NOT a peer-reviewed journal, and the paper that you cite is an opinion piece, not a description of research. It seems to me that at least one of your citations 8 - 12 must say this - any reason you are not using one or more of those?
Line 65 - 66 A citation to the Gulf Cooperation Council would be good so that the reader can determine which countries belong to this.
Line 90 - calorie not calory
Line 96 - supply from abroad, not supply for abroad
line 142 "A total of nine vegetables were chosen as target vegetables: these nine - not these nice.
Lines 147 - 153 - It would be helpful to have a definition of "local". Within how many km did you consider crop production to be local?
Lines 188 - 201. This section is still confusing to me. If you are calculating N and P using the 36 vegetables included in the supplemental table, how can you compare those 36 to the 9 vegetables included in the scenario calculations?
lines 199 - 201 suggested wording change to clarify: As our baseline we used 2018 data to estimate the N and P footprints of 36 vegetables; we identified nine vegetables that have high consumption rates in the prefecture and developed scenarios (described below) assuming that various percentages of those vegetables were grown using VF rather than conventional farming.
lines 197 - 199 - " Three scenarios were developed with a focus on the nine vegetable crops with imported ratios higher than the average imported ratios of the 36 vegetable crops in 2018." What exactly is being compared here? At one point you state that you are comparing 9 vegetables that were grown using VF - are these the same ones that you say have higher import ratios?
lines 203-204 suggest changing wording to "with an adjustment for the differences in chemical nature of N and P, explained after equation 5."
lines 209-210 - Your explanation of your equations is much improved. You use this term: "L is the loss of N in production [Mg N yr-1]" in your calculation of the one year N footprint of a given crop. Earlier in the paper you talk about the loss of N and P to the environment. Can you call L loss of N to the environment as you state earlier in the paper? As a reader, I wonder if there is a difference between loss of N to the environment and loss of N in production. If you are consistent with your terms, it is clearer.
Lines 205 - 224 - The equations make the assumption that burning is the only circumstance under which N is volatilized and lost "to the environment." This is not true, and you have a short discussion of other ways N volatilizes in your Discussion. Volatilization of N in the western US ranges from 12% to 60% of total amounts applied, depending on circumstances, so volatilization can be a significant loss. You need to acknowledge your lack of including this in your equations as a limitation of this study. The only loss you are accounting for is burning and there are many other ways that N can be lost. Your method could overestimate the amount of N released through conventional farming by not taking this into consideration.
There is no discussion of whether any crops are planted in Japan that fix N - legumes, alfalfa. That would also be a potential source of excess N. Although it need not be part of your calculation, if such crops are planted you need to acknowledge this as another limitation of this study.
Also, I tried to access the citation for the data you used for your calculations of crop uptake and amounts of N and P in crop residue and since it is in Japanese, I cannot read the data. It appears that data is from 2015, however. You are comparing it to other data from 2018 and that needs to be acknowledged and the impact, if any, of the difference in years needs a short discussion.
Lines 259 - 260 "potentially VF grown (potentially grown in VF with a high risk of failure due to insufficient social and economic acceptance)."
This raises two questions for me: 1. On what data are you basing your statement that there is a high risk of failure due to insufficient social and economic acceptance? Do you have data or a citation for this assertion? 2. Are there other places that are currently growing pumpkin or melon using VF? That would be important information, as most of the plants I am aware are grown in VF do not include these two. I am aware that people use trellises to grow pumpkins and melons in order to reduce ground space, but I have not heard of them using nutrient solution to replace soil growing these plants. If they are being grown in nutrient solution, please indicate where.
Lines 345 - 346 - "N2O emissions in agriculture are significant due to the utilization of fertilizer [80]" and use of manures - particularly when both are used. Both fertilizer and manure contribute to N2O emissions.
lines 406 - 407 "According to the survey results, which were discussed in 2.1.1, the post-disaster promotion of VF was revealed." There is no discussion of a survey in section 2.1.1 or any other section that I found. If the researchers conducted a survey, that should be disclosed in the paper and any protocols (IRB) if needed should also be discussed.
lines 412 - 414 "The risk posed by violent disasters to VF needs to be assessed and efforts to mitigate the potential damage need to be considered. In order to be disaster-proof, VF may require a back-up generator system, for example." If an earthquake reduces a building in which VF is taking place to rubble, a back-up generator will be of no help. I don't think there is a way to make anything, including VF, entirely disaster proof.
lines 453 - 454 "In the future, it is necessary to conduct additional footprint analyses by the life cycle assessment approach which including energy use, change in land use and transportation." Suggested wording change: In the future, this research can be expanded by conducting a life-cycle analysis of the environmental footprint and carbon emissions of VF compared to conventional agriculture including agricultural imports.
Author Response
Thank you very much again for your review and comments. We have revised the manuscript and responsed to reviewers' comments. Please see the attachment.

Reviewer 3 Report
I accept the corrected version.
Improving English still required.
Author Response
Thank you very much again for your review and many suggestions. We are glad our paper meets your approval and have responsed to reviewers' comments. Please see the attachment and find out the response at the end of file.
